# Mechanical Performance and Environmental Assessment of Sustainable Concrete Reinforced with Recycled End-of-Life Tyre Fibres

**DOI:** 10.3390/ma14020256

**Published:** 2021-01-06

**Authors:** Magdalena Pawelska-Mazur, Maria Kaszynska

**Affiliations:** 1Faculty of Civil and Environmental Engineering, Gdansk University of Technology, ul. Narutowicza 11/12, 80-233 Gdansk, Poland; 2Faculty of Civil and Environmental Engineering, West Pomeranian University of Technology in Szczecin, Al. Piastów 17, 70-310 Szczecin, Poland

**Keywords:** fibre-reinforced concrete, recycled steel fibres, micro-computed tomography, scanning electron microscopy, tensile strength

## Abstract

The presented research’s main objective was to develop the solution to the global problem of using steel waste obtained during rubber recovery during the tire recycling. A detailed comparative analysis of mechanical and physical features of the concrete composite with the addition of recycled steel fibres (RSF) in relation to the steel fibre concrete commonly used for industrial floors was conducted. A study was carried out using micro-computed tomography and the scanning electron microscope to determine the fibres’ characteristics, incl. the EDS spectrum. In order to designate the full performance of the physical and mechanical features of the novel composite, a wide range of tests was performed with particular emphasis on the determination of the tensile strength of the composite. This parameter appointed by tensile strength testing for splitting, residual tensile strength test (3-point test), and a wedge splitting test (WST), demonstrated the increase of tensile strength (vs unmodified concrete) by 43%, 30%, and 70% relevantly to the method. The indication of the reinforced composite’s fracture characteristics using the digital image correlation (DIC) method allowed to illustrate the map of deformation of the samples during WST. The novel composite was tested in reference to the circular economy concept and showed 31.3% lower energy consumption and 30.8% lower CO_2_ emissions than a commonly used fibre concrete.

## 1. Introduction

Due to the increasing EU car production, the problem of end-of-life tyre utilisation in an eco-friendly way is still growing. It is estimated that around 3.3 million tons of used tyres per year in Europe require recycling. In EC countries, restrictive regulations have been introduced to support solutions for the recovery, recycling or reuse of tyres towards the reduction of their harmful impact on the environment. In accordance with the Landfill Directive (1999/31/EC), the disposal of end-of-life tyres to landfills is prohibited. The directive introduced a ban on the storage of all used tyres since 2003, and since 2006 this ban included additionally crumbled tyres. Another End-of-Life Vehicle Directive (2000/53/EC) ordered the removal of tyres from vehicles before scrapping, and the Waste Incineration directive obliged cement plants using tyres as fuel to achieve lower limits for the content of pollutants in waste gases.

Used tyres (end-of-life tyres ELT) are recycled in two ways: towards obtaining energy or material. Novel solutions for raw materials or innovative products resulting from ELT are desired. However, still, up to 64% of used tyres are subject to energy recycling, i.e., they are burned mostly in cement kilns [1].

Due to the growing awareness of the need for sustainable development in the construction sector, new research has been conducted on the use of raw materials derived from tyre recycling processes. Tyre shredding in an ambient method is the most commonly used mode of ELT recycling. Pre-cut tyres are shredded in special granulators and rolling mills. Pneumatic separation is used to remove the textile fibres, and the steel fibres are removed by means of an electromagnet. Tyres are 100% recyclable. All tyre components, i.e., rubber, metals, and textiles, can be recovered. However, only rubber is widely reused in various commercial products, as evidenced by numerous studies published in the literature [2,3,4,5,6,7,8,9,10,11,12,13,14,15]. Other derived raw materials like steel and synthetic fibres have not yet found any wide application as a full-value raw material or a full-fledged component of other products. Tyres contain 16–25% of steel by mass. The separated recycled steel fibres (RSF) are treated as waste in the recovery of rubber. Their commercial value refers to the scrap price, and they are pressed into briquettes as separated steel fibres and sold to steel mills.

The use of steel fibres is much more difficult compared to the rubber raw material due to their heterogeneity, irregular shape, and frequent rubber contamination. In the majority of tyre recovery plants using the ambient grinding method, steel collected from separators is contaminated with rubber by up to 10%. Together with irregular geometric characteristics, these are the key features that determine the suitability of recycled fibres for their further high-quality applications. Fibres from various tyre recycling plants were considered for further research, but only those obtained from the recycling process invented by KAHL Holding GmbH showed the lowest degree of pollution. It was noticed that the latest KAHL technology focused on the high effectiveness of rubber recovery, allowing obtaining steel fibres with no greater than 2% impurity (Figure 1a).

Research on the methods of the valuable ELT steel application in the construction sector was conducted at numerous institutes. The majority of the studies focused on fibre concrete as one of the possible areas for the use of steel fibres from recycled tyres. According to numerous publications [16,17,18], the addition of uniformly dispersed reinforcement in the form of steel fibres improves the cooperation of fibres with the concrete matrix when transferring loads, reduces the tendency to form micro-shrinkage pattern in concrete and hinders the propagation of scratches when loading concrete. Steel fibres added to the concrete matrix, considered as a discrete reinforcement of concrete, prevent damage to the element after a crack.

Strengthening the concrete matrix with heterogeneous RSF fibres made of high-quality steel can improve the concrete properties. However, in most cases, the researchers had encountered a problem of high rubber contamination and developed a pretreatment method to clean fibres before their use in concrete [19,20,21]. The use of almost pure RSF fibres allows their effective use in concrete without any pretreatment processes.

The main objective of the research was to develop the method of valuable steel tyre cord application in the construction sector. This paper demonstrates the results of studies conducted in order to prove the feasibility of replacing industrially produced fibres (ISF) with RSF fibres in the concrete composite commonly used for industrial floors.

## 2. Materials

### 2.1. Components of Concrete Mix

In order to determine the full characteristics of the designed composite with RSF fibres, 3 different concrete mixes were designed based on the components previously selected within preliminary tests and recommendations for the design of industrial concrete floors.

CEM II/B-S 42.5N-NA low-alkali slag Portland cement (Gorazdze cement plant—Heidelberg Cement Group, Gorażdże, Poland) with an average specific surface area of 4110 cm^2^/g was used as the most commonly used cement in Poland for industrial floors. A PCE polymer superplasticiser (Sika, Warsaw, Poland) was used added at an amount of 1%.

Glacial origin aggregates found in Pomerania (Dech-Pol aggregates plant, Gdynia, Poland) were used to make all concrete mixtures. Natural washed aggregates with fractions 0/2, 2/8, and 8/16 were used. A sand point equal to 43% was assumed. For gravel, the percentage advantage of natural aggregate fraction 2/8 (60%) over the 8/16 (40%) mm fraction was adopted, as it is a solution reducing the cost of fibre concrete and supporting environmental protection due to the prevalence of fine aggregates occurrence in northern Poland.

### 2.2. Characteristics of Recycled Steel Fibres (RSF) and Industrial Steel Fibres (ISF)

The use of almost pure fibres RSF without any pretreatment was essential in research. RSF fibres (Figure 1a) are characterised by different shapes, lengths and diameters. Therefore, for a more detailed analysis of the geometric characteristics of RSF, an in-depth study was carried out using the scanning electron microscope methods including the Energy Dispersive X-ray Spectroscopy (EDS) spectrum (Jeol Ltd., Tokyo, Japan) and the micro-computed tomography (μCT) (Bruker, Kontich, Belgium).

The structural and chemical analysis of RSF fibres was performed using a Field Emission Gun (FEG) scanning electron microscope (Jeol Ltd., Tokyo, Japan), the operation of which is based on Energy Dispersive X-ray Spectroscopy (EDS) analysis. Use of a scanning microscope (model JSM-7900F, Jeol Ltd., Tokyo, Japan) combines the highest quality of imaging with chemical and structural analysis in the nanometer scale. EDS analysis allowed for the surface and volume identification of chemical elements included in the composition of tested RSF fibres.

SEM images (Figure 2) taken by use of a scanning electron microscope show that the surface of RSF fibres can be contaminated with rubber to various degrees. In addition, surface distortions are visible as a result of mechanical damages during use or the recycling process.

The EDS analysis showed that the chemical composition of some RSF fibres contains percentages of copper and zinc (Figure 3). The presence of both elements in approximately 2:1 weight ratio confirms that the steel cord fibres are covered with a thin layer of brass. Covering the fibres with a soft alloy coating is beneficial because of the further increase in the adhesion of fibres to the cement matrix.

The 3D Skyscan 1173 X-ray microtomography (Bruker, Kontich, Belgium) was used to perform a comprehensive analysis of the characteristics and distribution of fibres in the concrete. X-ray microtomography (μCT) is a 3D imaging technique that uses X-rays to create cross sections of a physical object to recreate a virtual model (3D model) without destroying the sample. The following parameters were determined as part of the research:the percentage distribution of the fibre length,percentage distribution of the diameters of the fibres,percentage distribution of fibre slenderness,percentage distribution of the fibre orientation.

The research was carried out on scans of 75 mm side samples made of RSFRC concrete. The maximum size of the test samples was narrowed down to a cube with a side of 70 mm to eliminate boundary disturbances. As part of the analysis of the characteristics of fibres, the following parameters were determined: relative frequency of the fibre length and relative frequency of the fibre diameter.

The study has proved that about 77% of RSF fibres have a length in the range of 5 to 30 mm (Figure 4). When analysing the fibre length distribution, it should be emphasised that it concerns real lengths. RSF fibres, due to their irregular shape, have a much smaller effective length *l_ef_* of anchoring the fibres. At the same time, it was found that about 90% of RSF fibres have a diameter in the range of 0.1 to 0.4 mm (Figure 5). Another calculation of the average percentage slenderness distribution of RSF fibres indicated that the slenderness of the fibres is in the range 10–150, and the most (~27%) has the slenderness in the range 30–60 (Figure 6). The geometrical characteristics of the steel fibres are illustrated in Figure 1a with a visible variable length and irregular shape of the fibres. RSF fibres are made of steel with a tensile strength of over 2200 MPa [22], which is twice as high as the commonly used fibres for concrete. Due to the low fibre contamination, a density of 7800 kg/m^3^ was assumed for RSF. The mechanical and geometrical features of RSF are presented in Table 1.

For the comparative tests, industrially produced steel fibres (ISF) (Figure 1b) with a hook anchorage, 0.5 mm in cross section, 25 mm long and specified by the manufacturer (Baubach Metall E.S. GmbH, Effelder, Germany) as WLS-25/0.5/H were used. According to the technical characteristics declared by the manufacturer (Table 2), the tensile strength of the fibres is at least 1100 N/mm^2^. The above-mentioned fibres, in their length and cross section, are the closest to those obtained in the ambient recycling by the KAHL processing.

### 2.3. Composition of Concrete Mix

Concerning the determination of the physical and mechanical characteristics of the designed composite, a wide range of tests was carried out on unreinforced concrete (NC), reinforced with ISF fibres (SFRC) and RSF fibres (RSFRC). The composition of the concrete mix was determined in accordance with the recommendations for the design of industrial concrete floors and based on the conclusions of preliminary tests.

The recipe of each mix (Table 3) was initially appointed analytically, based on the method of three equations, assuming that the amount of cement in the mix is 320 kg/m^3^ and the factor *w*/*c* = 0.5. The additional design assumptions were as follows; S3 consistency class of concrete mix, concrete class of min. C25/30; adoption of RSF fibres with a content of 50 kg/m^3^ and a comparatively ISF fibres with a content of 25 kg/m^3^. The application of double content of RSF fibres versus ISF fibres was conducted due to previous study on geometrical characteristics of the recycled fibres, where it was demonstrated that only about 60% of RSF fibres improve the parameters of the composite.

## 3. Methodology

### 3.1. Assigning the Rheological Properties of the Concrete Mix

The study was initiated by assigning the rheological properties of the concrete mix. The research was conducted on the determination of the consistency, the air content and density of concrete mixes which contained fibres (ISF and RSF) or were plain. The consistency of the concrete mix was tested by the Abrams cone slump method in accordance with the EN 12350-2 standard, the air content was determined using the pressure method based on the Boyle–Mariott law in accordance with the EN 12350-7 code and the concrete mix density was tested in accordance with the EN 12350-6 standard.

### 3.2. Tests on Physical and Mechanical Features of Concrete Composites

The next stage of the research was concerned with the determination of the parameters of concrete composites. It contained the analysis of the distribution of ISF and RSF fibres in studied concrete mixes. The imagines of the distribution of ISF and RSF fibres in the samples of concrete composites were taken, and the orientation of their position in concrete was determined in relation to the Cartesian system. The research was carried out on 3D scans of 75 mm side samples made of RSFRC and SFRC concrete (Figure 7). The experimental procedure followed the research on the analysis of the RSF characteristics (Section 2.2), where 3D Skyscan 1173 X-ray microtomography was used.

In order to designate the full characteristics of the physical and mechanical features of the proposed new composite, a wide range of tests was performed, starting with the compressive strength test. That test was carried out on 6 cubic samples with a side of 150 mm for each of the 3 series, in accordance with the EN 12390-3 standard, using CONTROLS testing machine (Controls S.p.A., Liscate, Italy) enabling electronic recording of the results.

Assigning other mechanical properties, particular emphasis was put on the determination of the tensile strength of the composite. This parameter was appointed by three different methods: tensile strength testing for splitting (Brazilian method), residual tensile strength test (3-point test) and testing of tensile strength by a wedge splitting (WST method). The last two methods allowed us to determine the dependence of *strength-CMOD* for the tested samples.

The Brazilian method of splitting a concrete sample during compression is the most commonly used method of testing concrete tensile strength. According to the EN 12390-6 standard, tests were performed on cubic samples with a side of 150 mm, 6 samples for each of the 3 series after 28 days of maturation, using CONTROLS testing machine.

In the case of composites reinforced with steel fibres, the determination of the residual tensile strength is the key test used in the calculation of structural elements made of fibre concrete according to the recommendations of RILEM TC 162-TDF. The addition of steel fibres is intended to transfer stresses in the tension zone after the first crack has appeared. Tensile strength was measured in accordance with EN 14651 code. The method is performing a 3-point test (Figure 8) in order to determine the proportionality limit and residual strength on beam-shaped samples with dimensions: 150 × 150 × 700 mm (3 samples for each concrete mix: NC, SFRC and RSFRC). The samples were prismatically incised in the middle of the span to the height of 25 mm in order to force a crack in the beam axis. The span of beam supports was 500 mm.

The wedge splitting test (WST) is an alternative method of determining the σ-CMOD relationship. The test was developed in 1986 by H.N. Linsbauer and E.K. Tschegga [23]. Since then, it has been widely used to determine the mechanical properties of fractures of brittle and quasi-brittle materials. The optical method of digital image correlation (DIC) was used to determine the map of sample deformation during stretching by splitting with a wedge. The applied DIC system was equipped with a camera enabling obtaining two-dimensional results. As part of this work, the WST tests were carried out on cubic samples with a side of 71 mm made of three different concrete formulas: without fibres—NC, concrete reinforced with ISF fibres—SFRC, and RSF—RSFRC fibres. The samples were incised in the upper plane to a depth of 10 mm and a width of 5 mm in the centre of the sample, where the relationship between the crack width (CMOD) and the applied force F was then tested. The samples were loaded with a controlled movement speed of 0.001 mm/min in a testing machine INSTRON 5569 (Instron, Electromechanical & Industrial Products Group, Norwood, MA, USA) (Figure 9). This type of control allowed us to obtain a constant decrease in strength in the period after reaching the maximum stress corresponding to the applied force F_max_.

Due to the intended use of the designed composites in industrial concrete floors, additional tests were performed on the impact of RSF fibre addition on adhesion and abrasion of concrete. The adhesion of concrete layers was determined by the pull-off test in accordance with EN 1542 code. The test was performed for 7 samples from each of the 3 series of concrete. The abrasion resistance was determined using a Boehme dial, over which a test specimen with dimensions 71 × 71 × 71 mm^3^ was fixed, according to the procedure described in EN 13892-3. Abrasion was determined by the loss of height and volume of the sample determined on the basis of the weight loss of the tested sample.

## 4. Results

### 4.1. Analysis of Fibre Distribution in a Cement Matrix

Concrete reinforced with fibre is considered an anisotropic material. Using 3D X-ray micro-computed tomography, images of the distribution of ISF and RSF fibres in the samples of concrete composites were taken, and the orientation of their position in concrete was determined in relation to the Cartesian system. The results of the comparison of the percentage distribution of the RSF and ISF fibres in relation to the *Z*-axis (Figure 10) prove the advantage of the fibres arranged in a perpendicular direction to the concreting direction of the samples. Approximately 73% RSF and 74% ISF are laid at an angle of Θ in the range of 45° to 90°. The observed phenomena, although contradicting the assumption of anisotropy of fibre-reinforced concrete, in the case of floor applications, are advantageous due to the predominance of tensile forces acting perpendicularly to the direction of concreting.

The analysis of the fibre inclination angle Φ in the XY plane carried out for the same samples showed an even distribution for both types of fibres. The percentage distribution of the position angle of the fibres Φ in the XY plane for 5° did not show a greater share than 4.2% for both ISF fibres and RSF fibres (Figure 11).

3D images of samples (Figure 12) show an even distribution of ISF and RSF fibres in concrete. An illustrated lack of clusters of fibres, i.e., “balls”, is a significant finding. It proves no additional requirements are needed for mixing the components during the production of the proposed fibre concrete.

### 4.2. Tests on the Rheological Properties of the Concrete Mix

Studies have shown that dosing of ISF-dispersed reinforcement in the amount of 25 kg/m^3^ caused the Abrams Cone slump by 6 cm, and the addition of RSF fibres in the amount of 50 kg/m^3^ resulted in the decrease by another 2 cm in comparison to mixtures without fibres (Figure 13). These results correspond to a lowering of the consistency class for both mixtures modified with steel fibres from S4 (for NC) to S3 (for SFRC and RSFRC).

During testing, efforts were made to avoid sedimentation of the ingredients while maintaining relatively low air content. The air content test results (Figure 14) show the increase in air content at concrete mixes modified with still fibres: up to 2.15% for the SFRC mix and up to 1.90% for the RSFRC mix versus 1.70% for unmodified concrete mixes (NC).

### 4.3. Compressive Strength of Concrete

The results of compressive strength tests are presented in Table 4 and Figure 15.

### 4.4. Tensile Strength of Concrete

The tensile strength of the composite was appointed by three different methods: tensile strength testing for splitting (Brazilian method), residual tensile strength test (3-point test) and testing of tensile strength by a wedge splitting (WST method). Results of these tests are presented in Table 5.

In the case of tensile strength at splitting, the average strength of ISF fibre modified concrete is 31% higher, and in the case of RSF fibre modified concrete, it is 43% higher than the average strength of reference concrete without fibres.

The 3-point bending test allowed to determine the dependence of the applied force on the deflection (*F-deflection*) and the tensile stresses on the opening width of crack (*σ-CMOD*) for composites without fibres and modified with ISF and RSF fibres with a crack width up to 4 mm. To better illustrate the behaviour of samples in the early deformation phase, Figure 16 compares plots of *F-deflection* relationships for non-fibre (NC) and fibre (SFRC and RSFRC) composites obtained during residual strength testing at a deflection of 0.4 mm, which shows differences in the behaviour of the samples before scratching.

The *σ-CMOD* relationship diagram (Figure 17) illustrates the convergence of results for samples containing ISF fibres and RSF fibres. According to the results obtained (Table 5), RSFRC composites demonstrate a higher residual flexural strength *f_R1k_* corresponding to a crack opening equal to *CMOD_1_* = 0.5 (*f_R1k_* = 2.45 MPa) versus *f_R1k_* = 2.29 MPa for SFRC composites. A similar relationship applies to the residual strength *f_R3k_* corresponding to a crack opening equal to *CMOD_3_* = 2.5. For RSFRC samples, the residual strength *f_R3k_* is 1.26 MPa, and for SFRC samples the residual strength *f_R3k_* is significantly lower, it is 0.91 MPa.

The tensile strength test by splitting referring to the WST method showed the strength *σ_NT_* of NC concrete constituted about 60–70% of the tensile strength by splitting of concrete reinforced with steel fibres (SRFC and RSFCR). The values of the corresponding maximum tensile stress *σ_NT_* for concrete: NC, SRFC and RSFCR are shown in Table 5. The diagrams of the *σ-CMOD* relationship (Figure 18) demonstrate a very high convergence of results for samples reinforced with diffuse ISF and RSF fibres.

The optical method of Digital Image Correlation (DIC) was used to determine the map of sample deformation during stretching by wedge splitting test (WST). By analysing the places of deformation concentration, the sample cracking zone was located. Comparison of deformation propagation map images in the elastic zone (for a point corresponding to the application of ~85% force F_max_ behind the vertex (85% after F) indicates much faster crack penetration in concrete without fibres. In addition, it was observed that the crack location zone was more curved in the case of concrete with ISF fibres than RSF fibres (Figure 19). The colour scale indicates the intensity of the horizontal deformation of the RSFRC sample.

The above observation of the WST test confirmed the picture showing the scratches formed in the ISF sample (Figure 20).

### 4.5. Adhesion and Abrasion of Concrete Tests

The results of a pull-off peel test are demonstrated at Table 6, while the results of Boehme’s abrasion resistance tests are shown at Table 7.

### 4.6. Comparative Analysis of Energy Consumption and CO_2_ Emissions of the Production of Concrete Reinforced with ISF and RSF Fibres

To determine the energy consumption and CO_2_ emission parameters for individual components of fibre concrete, indicators published by the Central Statistical Office for the Polish Economy [24] (Table 8) and indicators derived from scientific studies were adopted [24].

## 5. Discussion

### 5.1. Impact of RSF Fibre Addition on the Rheological Properties of the Concrete Mix

The addition of steel fibres significantly affects all rheological properties of the fresh concrete mix. Fibre parameters such as type, length, shape and their content in the cement matrix impact on workability, consistency and air content. The results obtained so far relating to concretes modified with steel fibres prove that workability and consistency deteriorate with the increase in steel fibre content. The above phenomenon is explained by the fact that the spatially condensed fibre system in the mixture hinders the free movement of aggregate grains. In addition, part of the cement paste directly surrounds fibres instead of sand grains.

Research conducted on fibre concrete in the 1970s [16] showed that the consistency of concrete modified with RSF fibres depends on two parameters: fibre content in the concrete mix and their slenderness *l/d*. The results obtained on consistency tests are consistent in the abovementioned statement through lowering the consistency class for both mixtures modified with steel fibres from S4 to S3.

The air content in the concrete mix is closely related to the compacting process and the occurring phenomenon of removing air from the cement paste. In case of fibre concrete mixes, the reduction of air bubbles during compacting is more difficult due to higher aeration of the mix, and the limitation of components displacement due to the fibre addition. If the compacting process is too long, a risk appears of disturbing the spatial distribution of the fibres and, consequently, their falling to the bottom of the mould. The test results (Figure 14) confirmed the effect of fibres on the increase in air content: up to 2.15% for the SFRC mix and up to 1.90% for the RSFRC mix. Increased air content in fibre reinforced concrete was also observed when testing hardened concrete samples with an X-ray micro-computed tomography (Figure 9), where SFRC concrete contained 2.76% and RSFRC concrete only 1.73% of air voids. It has been shown that the effect of fibres on loosening the mix structure and hardened concrete is greater in the case of homogeneous fibres (ISF) than hybrid fibres (RSF).

### 5.2. Impact of RSF Fibre Addition on Strength Parameters of the Designed Composite

According to the literature [26], experimental tests of compressive strength show no significant impact of dispersed reinforcement with a content of up to 1% of concrete volume. It was observed that an addition of ISF fibres in an amount exceeding 1.5% effect an increase in compressive strength up to 15%. Other sources [27,28] indicate an increase of 10 to 30% in the compressive strength of fibre concrete already with the addition of ISF fibres in the amount exceeding 0.5% of the concrete volume. Due to the loosening of the fibre concrete structure at the presence of fibres, experimental tests may also show a slight decrease in strength.

Test results cited in the literature on fibre concrete made from recycled tyres showed no significant effect on compressive strength [29,30] or a slight increase in strength (by 9% at a fibre content of 0.41% and by 17% at a volumetric content fibre 0.46%).

Own research showed an increase in compressive strength (Table 4 and Figure 15) by 13.5% for concrete modified with ISF fibres (at 0.32% fibre volume content) and by 22% at RSF (at 0.64% fibre volume content). The high compressive strength obtained in the test for both concretes modified with steel fibres is explained by the inclusion of fibres in the control of static stress in the three-axis stress state occurring in compressed cubes. When the concrete expands in a direction perpendicular to the compressive force, the fibres can bridge emerging scratches until the steel fibres are torn out of the cement matrix, which results in an increase in compressive strength.

When determining the strength parameters, special emphasis was placed on the tensile strength of the new composite. Low tensile strength in elements of ordinary concrete has an effect on the formation of scratches. In published results of world studies [26,27] it has been shown that the addition of steel fibres increases the tensile strength of concrete by 10 to 30% with a spatial arrangement of fibres and from 30 to 70% with directed fibre distribution.

During that study, three different tests on the tensile strength of the composite: splitting test (Brazilian method), residual tensile strength test (3-point test) and a wedge splitting test (WST method) have confirmed the increase in tensile strength when modifying concrete with RSF fibres (Table 5).

Analysis of the graphs of the *F-deflection* relationships at the bending test of fibre concrete elements (Figure 16) demonstrated that the formation of the first crack does not lead to the sudden destruction of the element made of RSFRC concrete. It is assumed that the development of a critical crack results in tensile stresses to be absorbed by adjacent fibres. Thus, subsequent scratches lead to further deformation of the element but not to complete destruction.

The results of experimental studies also showed a significant impact of sample size on concrete tensile strength when determined by various methods: bending, splitting and direct stretching. It was found that the value of concrete tensile strength at bending is greater than determined in the splitting test. The assumption of linear elasticity and flat sections of concrete when determining the tensile strength at bending causes an increase in the calculated value of tensile strength compared to the actual value.

Analysis of the results of their own research confirms the above observation. The determined value of concrete tensile strength at bending *f_L_* obtained during the 3-point test is higher by 21% for RSFRC and by 36% for SFRC than the strength *f_ct,sp_* determined in the Brazilian splitting test. Similarly, the effect of sample size, a different method of force application, and a calculation procedure based only on the effect of the horizontal component when calculating the size may explain the much lower tensile strength value obtained by splitting according to the WST method than in the case of the Brazilian method. Despite the differences in values of the tensile strengths, an increase in tensile strength for RSFRC concrete from 30% (tensile strength determined in the bending test) to 70% (tensile strength determined in the splitting test during the WST test) relative to NC concrete was observed for all measuring methods (Figure 21).

Additional analysis of the *σ-CMOD* charts obtained during the 3-point test and the WST test (Figure 22) indicates a great convergence of the nature of the behaviour of the samples with the RSF and ISF fibres in the elastic zone, which indicates the possibility of replacing ISF fibres with recycled RSF fibres without reducing the strength parameters fibre concrete.

### 5.3. Impact of RSF Fibre Addition on Adhesion and Abrasion of Concrete

In the case of steel fibre-modified composites, in the case of a pull-off peel test, the dispersion of results is quite large due to the small peel surface and the heterogeneous distribution of the fibres in its area. No apparent effect of steel fibres on the adhesion in concrete was observed. The average surface tensile strength for a composite without NC fibres and for a composite modified with ISF fibres is 2.50 MPa and 2.45 MPa, respectively. The highest peel strength (2.49 MPa) was observed for the composite modified with RSF fibres. The test results confirm that the designed RSFRC concrete meets the requirement for industrial floors, where it is recommended that the value of the surface tensile strength tested by the pull-off test be above 1.5 MPa (Table 6).

The results of tests on concrete abrasion resistance showed a significant influence of steel fibres on composite abrasion resistance. The test carried out on the Boehme disk showed a volume loss for the SFRC samples smaller by 11%, and for the RSFRC samples smaller by 15% compared to those obtained for NC concrete (Table 7).

### 5.4. Environmental Impact of the Production of Concrete Reinforced with ISF and RSF Fibres

Comparing the energy consumption and CO_2_ emission for the concrete recipe with RSF fibres (Table 9) to ISF fibres (Table 8) shows 31.3% lower energy consumption and 30.8% lower CO_2_ emission than for concrete with ISF fibres. It is clearly demonstrated that industrially produced steel fibres, due to complicated and energy-intensive production processes, are the second component (after cement) significantly affecting the natural environment. Replacing industrially produced steel fibres with waste fibres obtained in the recovery of rubber from used tyres may impact on the reduction of energy consumption and greenhouse gas emissions. Considering the extension of reused components in tyre recycling process, the proposed solution fully responds to the Circular Economy concept.

## 6. Conclusions

In this study, the possibility of replacing industrial fibres in concrete with RSF fibres treated as waste from tyre recycling processes was demonstrated. To estimate the efficiency of RSF fibre concrete, a wide range of strength tests was performed, emphasising the determination of the tensile strength of the composite. The research and analysis showed that RSF fibres have parameters comparable to those of industrial steel fibres used to reinforce concrete industrial floors.

The following conclusions can be drawn from the research described in this paper.

The geometric examination of the fibres confirmed that the fibres from the recycling of tyres used for the tests are characterised by varying lengths, diameter, slenderness and shape. Such a mixture can be classified as hybrid fibres, which show higher efficiency in concrete than fibres of the equal length. However, considering the geometrical characteristics of the fibres, is was shown that only about 60% of the RSF fibres in concrete improve the parameters of the composite. This indicates the application of double content of RSF fibres (50 kg/m^3^) versus ISF fibres (25 kg/m^3^) in proposed concrete mix.The research showed that steel fibres’ addition significantly affects all rheological and mechanical properties of the concrete: its workability (lower), consistency (reduction by one class), air content in the mix, and strength parameters. The addition of steel fibres can enhance concrete performance, especially the compressive strength (by 13.5% for composite modified with ISF fibres and by 22% for concrete modified with RSF fibres). Tensile strength tests carried out by three methods: Brazilian splitting, bending (3-point test), and WST splitting confirmed the increase in tensile strength when modifying concrete with RSF fibres, respectively, by 43%, 30% and 70% in comparison to the average strength of reference concrete without fibres. Moreover, RSF fibres significantly improved the abrasion resistance of the composite (by 42%).The calculation of environmental parameters of concrete with RSF fibres showed significantly lower energy consumption (by 31.3%) and lower CO2 emission (by 30.8%) than concrete with ISF fibres due to the energy-consuming production processes of industrial fibres.

The main limitation for the substitution of ISF fibres with fibres derived from the recycling of tyres in concrete was recognised in improper distribution of irregular RSF fibres and the threat of fibre clusters, i.e., “balls”. However, the 3D images of samples had demonstrated the lack of clusters of RSF fibres and therefore proved that no additional requirements are needed for mixing the components during the production of proposed fibre concrete.

The above-mentioned conclusions prove the possibility of replacing industrial fibres with RSF fibres without reducing fibre concrete’s strength parameters. In addition, due to the growing public awareness of the need to protect the environment and the climate consequences associated with excessive greenhouse gas emissions, all solutions to reduce CO_2_ emissions and waste recycling have great potential for rapid commercialisation. Therefore, further investigations are recommended to explore the possibility of utilising RSF as an eco-friendly material to mitigate the challenges of the sustainable construction industry.

## Figures and Tables

**Figure 1 materials-14-00256-f001:**
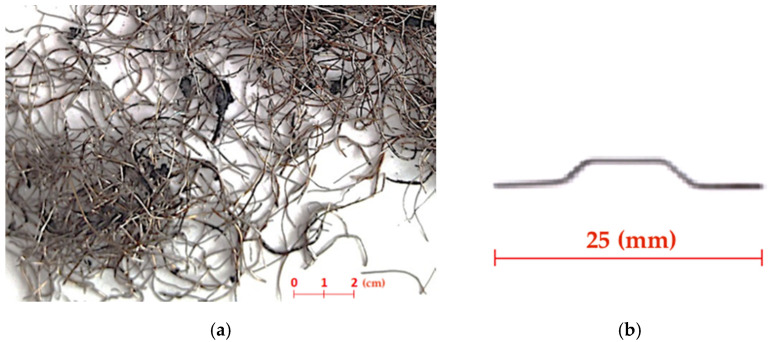
(**a**) Recycled steel fibres (RSF) derived in KAHL technology (ambient grinding method). (**b**) Industrial produced steel fibres (ISF).

**Figure 2 materials-14-00256-f002:**
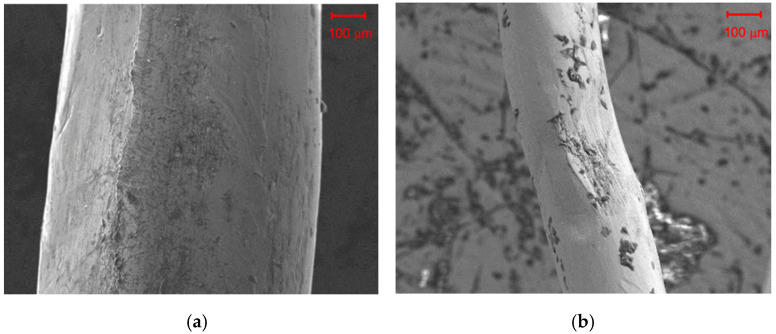
SEM image by use of a scanning electron microscope image of RSF fibres with a diameter of: (**a**) 600 μm and (**b**) 200 μm.

**Figure 3 materials-14-00256-f003:**
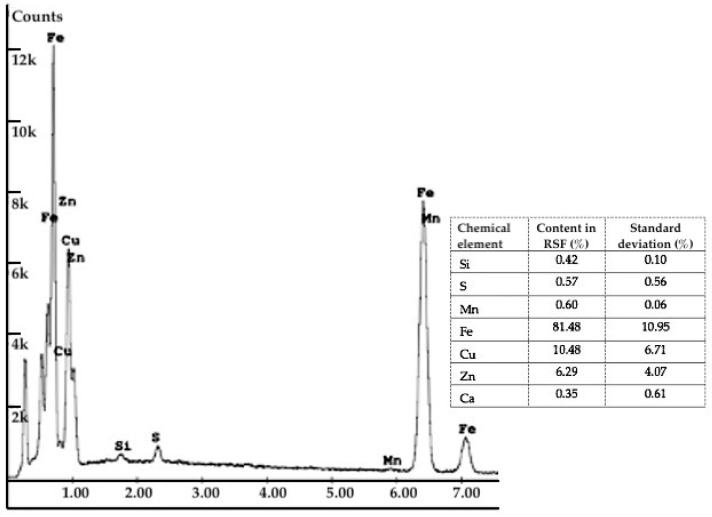
EDS spectrum for RSF recognised under X-ray analysis.

**Figure 4 materials-14-00256-f004:**
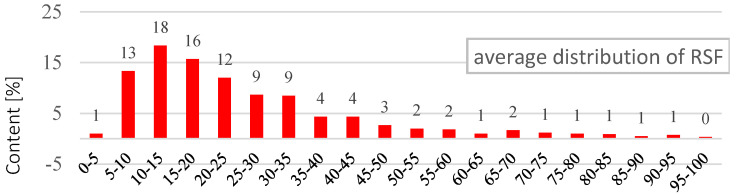
The average percentage of RSF length distribution.

**Figure 5 materials-14-00256-f005:**
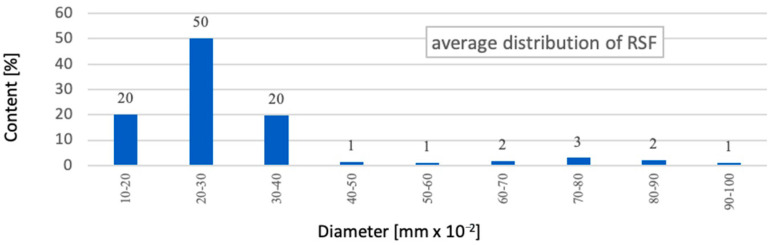
The average percentage of RSF diameter distribution.

**Figure 6 materials-14-00256-f006:**
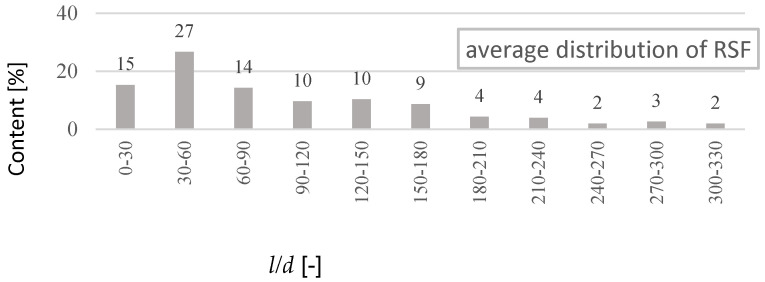
Average percentage RSF slenderness (*l/d*) distribution.

**Figure 7 materials-14-00256-f007:**
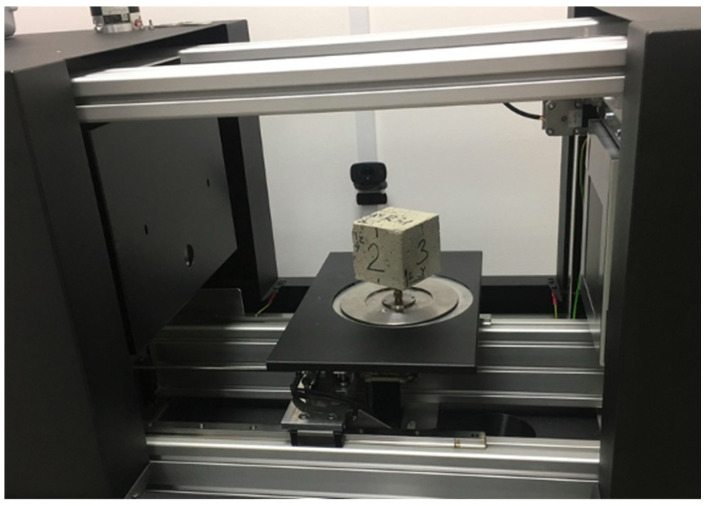
Determination of the distribution of ISF and RSF fibres in the samples of concrete composites using 3D Skyscan 1173 X-ray microtomography.

**Figure 8 materials-14-00256-f008:**
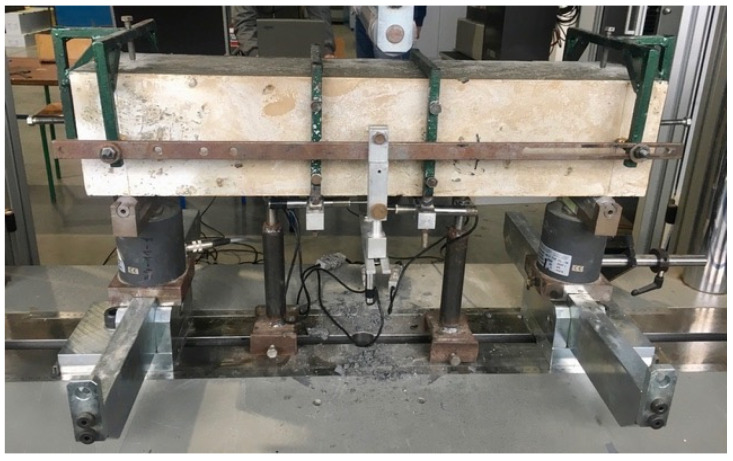
Test stand for determination of the residual tensile strength (3-point bending test).

**Figure 9 materials-14-00256-f009:**
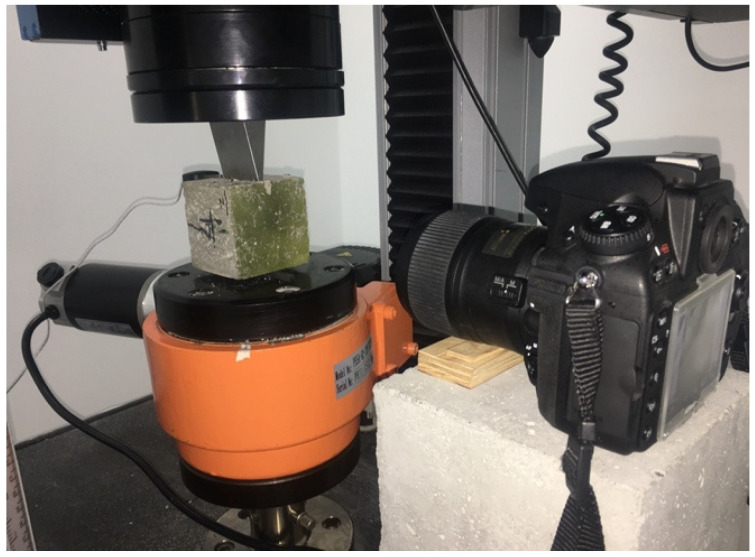
Test stand for testing tensile cracking characteristics when splitting concrete sample with a wedge (WSP) using digital image correlation (DIC) method.

**Figure 10 materials-14-00256-f010:**
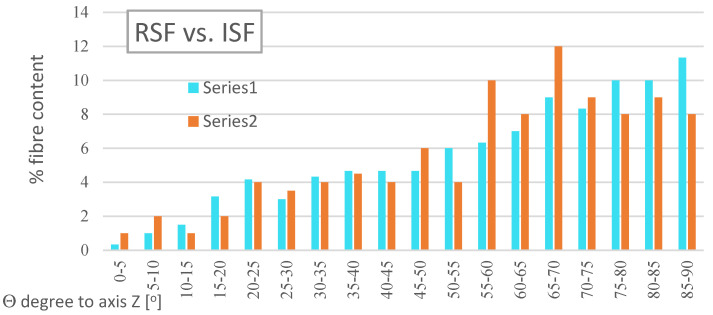
The average percentage fibre distribution in samples containing RSF fibres (Series 1) and ISF fibres (Series 2) in relation to their location, angle Θ, relative to the *Z*-axis.

**Figure 11 materials-14-00256-f011:**
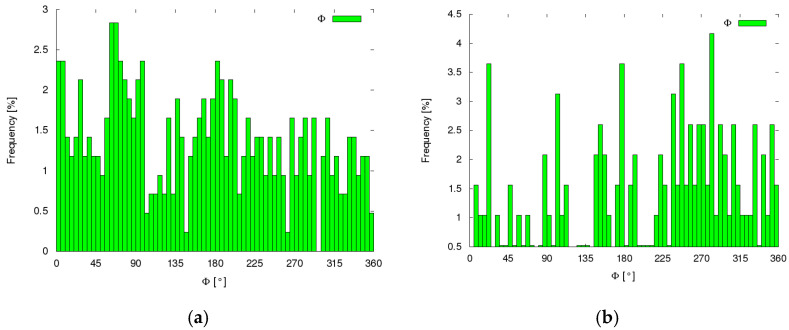
The percentage distribution of fibres in relation to their position in the XY plane (Φ angle): (**a**) RSF fibres and (**b**) ISF fibres.

**Figure 12 materials-14-00256-f012:**
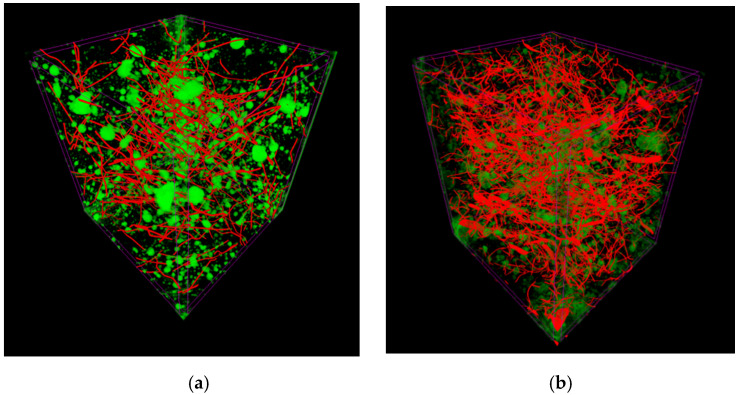
3D computer micro-tomography image of SFRC (**a**) and RSFRC (**b**); fibre distribution (red): SFRC—0.32% and RSFRC 0.64%; and air pores (green): SFRC—2.76% and RSFRC 1.73%.

**Figure 13 materials-14-00256-f013:**
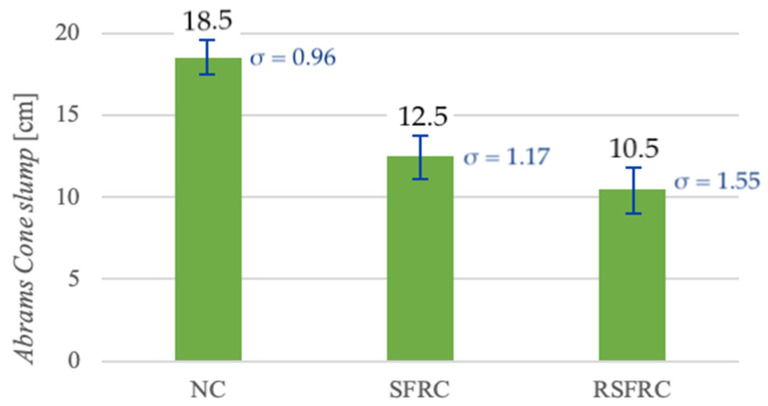
Comparison of the consistency test results with the cone slump test for mixtures without fibres (NC) and with steel fibres (SFRC and RSFRC).

**Figure 14 materials-14-00256-f014:**
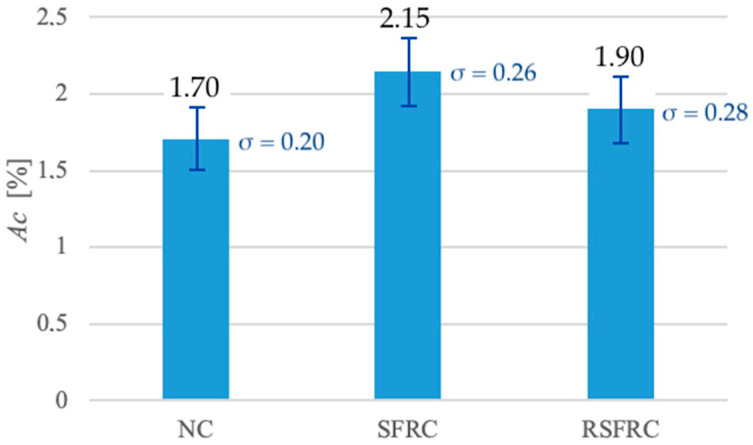
Comparison of the air content (*Ac*) for mixtures without fibres (NC) and with steel fibres (SFRC and RSFRC).

**Figure 15 materials-14-00256-f015:**
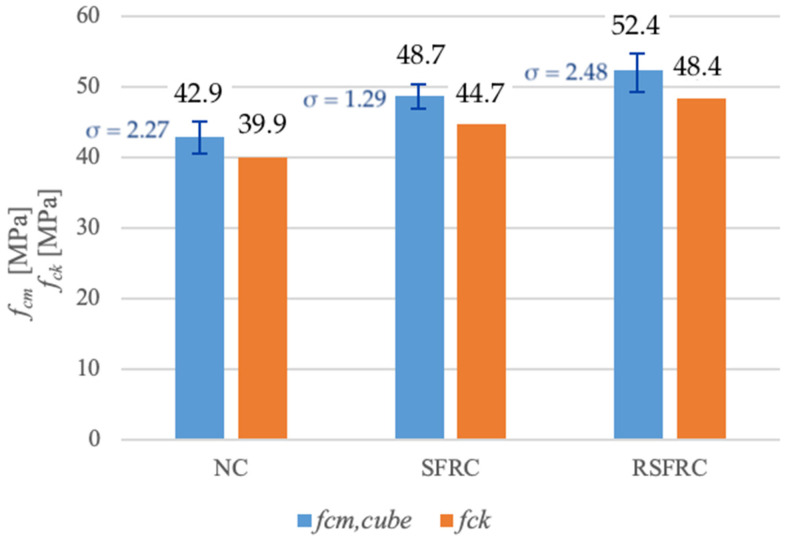
Comparison of medium (*f_cm_*) and characteristic (*f_ck_*) compression strength of tested concretes without fibres (NC) and with steel fibres (SFRC and RSFRC).

**Figure 16 materials-14-00256-f016:**
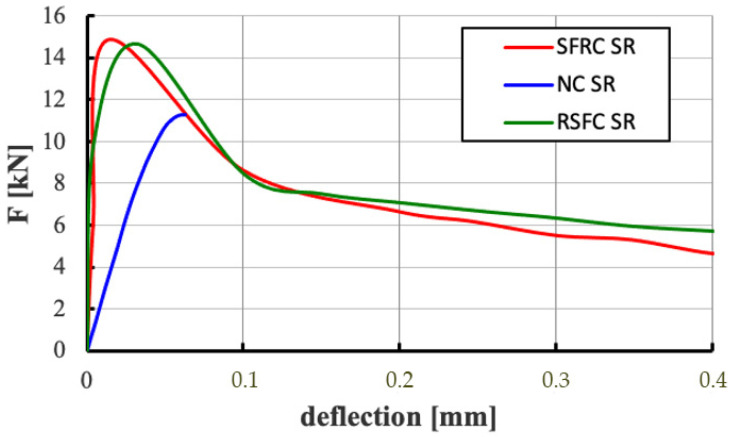
Comparison of diagrams of *F-deflection* relationships for non-fibre (NC) and fibre (SFRC and RSFRC) concretes obtained during the residual strength test with deflection up to 0.4 mm (3-point test).

**Figure 17 materials-14-00256-f017:**
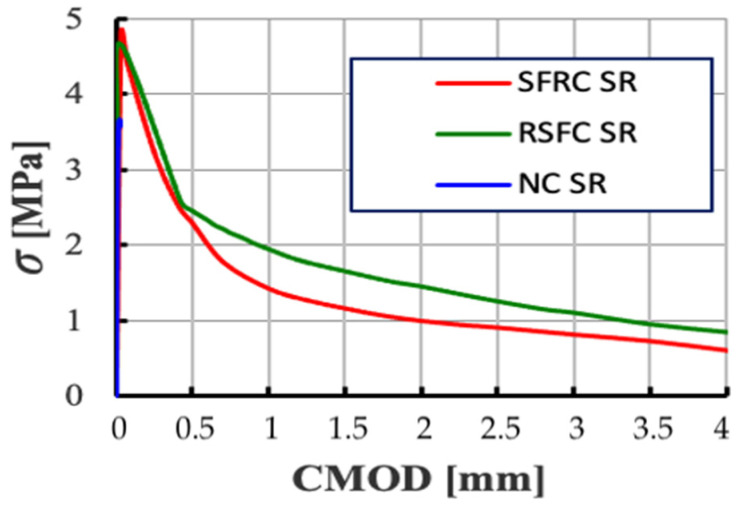
Comparison of diagrams of *σ-CMOD* relationships for concrete without fibres (NC) and with fibres (SFRC and RSFRC) obtained during the residual strength test for CMOD up to 4 mm (3-point test).

**Figure 18 materials-14-00256-f018:**
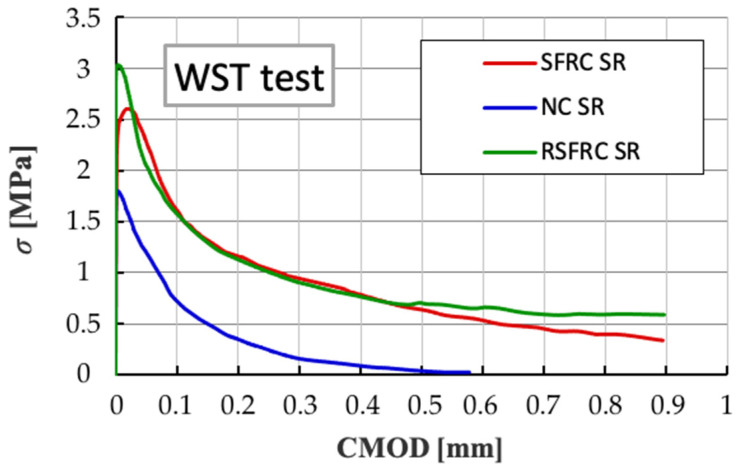
Comparison of diagrams of *σ-CMOD* relationships for concrete without fibres (NC) and with fibres (SFRC and RSFRC) at WST test.

**Figure 19 materials-14-00256-f019:**
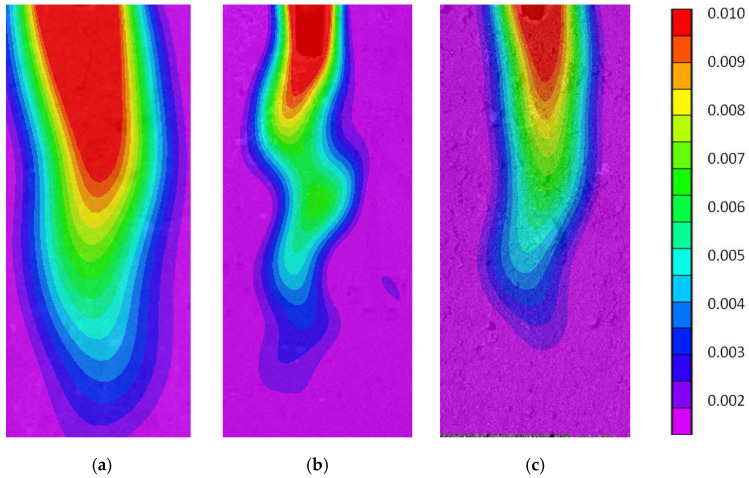
Images of deformation propagation map in the elastic zone obtained using Digital Image Correlation (DIC) for samples: (**a**) NC, (**b**) SFRC and (**c**) RSFRC.

**Figure 20 materials-14-00256-f020:**
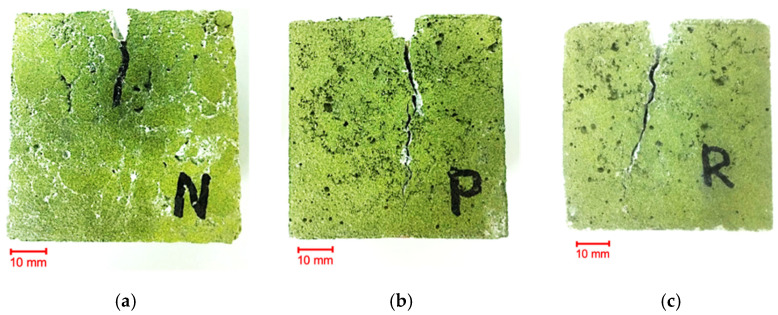
Pictures were taken with a digital camera illustrating the scratches formed in the samples as a result of the WSP test: (**a**) NC), (**b**) SFRC and (**c**) RSFRC.

**Figure 21 materials-14-00256-f021:**
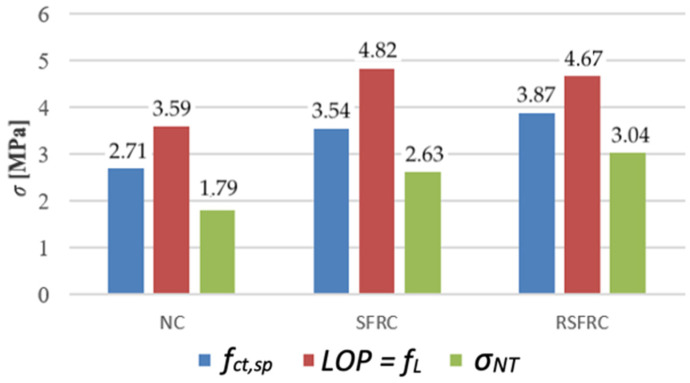
Comparison of tensile strength of tested concretes without fibres (NC) and with steel fibres (SFRC and RSFRC) appointed by Brazilian method, 3-point test and WST.

**Figure 22 materials-14-00256-f022:**
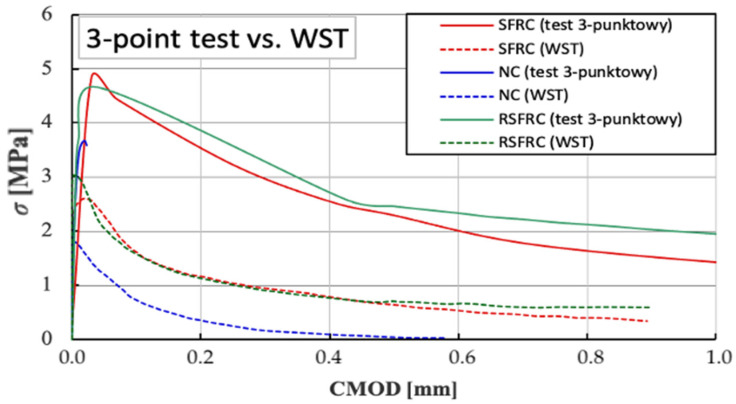
Comparison of *σ-CMOD* diagrams determined in the bending tests (3-point test) versus wedge splitting test (WST).

**Table 1 materials-14-00256-t001:** Mechanical and geometrical features of RSF.

Characteristics of RSF	Value
RSF length (mm) (77%)	5–30
RSF diameter (mm) (90%)	0.1–0.4
RSF slenderness	10–150
RSF figure	irregular
RSF tensile strength (MPa) [22]	2200
RSF density (kg/m^3^)	7800

**Table 2 materials-14-00256-t002:** Mechanical and geometrical features of ISF (according to the producer’s declaration).

Characteristics of ISF	Value
ISF length (mm) (77%)	25
ISF diameter (mm) (90%)	0.5
ISF slenderness	50
ISF Steel type	Group I EN 14889-1:2006
ISF tensile strength (MPa)	1100

**Table 3 materials-14-00256-t003:** Composition of concrete mixes without fibres (NC), with ISF fibres (SFRC), and with RSF fibres (RSFRC).

Concrete Mix Composition	NC	SFRC	RSFRC
cement CEM II/B-S 42.5N-NA (kg/m^3^)	320	320	320
sand 0–2 mm (kg/m^3^)	457	456	454
gravel 2–8 mm (kg/m^3^)	808	805	801
gravel 8–16 mm (kg/m^3^)	687	684	681
water (kg/m^3^)	160	160	160
ISF fibres WLS-25/0.5/H (kg/m^3^)	-	25	-
RSF fibres (kg/m^3^)	-	-	50
superplasticiser (kg/m^3^)	3.2	3.2	3.2

**Table 4 materials-14-00256-t004:** Compression strength results after 28 days.

Compression Strength	NC	SFRC	RSFRC
Compression force *F_c_* (kN]	966	1096	1179
medium compression strength *f_cm_* (MPa)	42.9	48.7	52.4
Standard deviation *(f_cm_)* (MPa)	2.27	1.29	2.48
characteristic compression strength *f_ck_* (MPa)	39.9	44.7	48.4
Class of Concrete	C30/37	C35/45	C35/45

**Table 5 materials-14-00256-t005:** Comparison of tensile strength for concretes without fibres (NC) and with fibres (SFRC and RSFRC).

-	NC	SFRC	RSFRC
**Tensile strength testing for splitting**	-	-	-
*f_ct,sp_* (MPa]	2.71	3.54	3.87
Increase to NC (%]	-	31	43
Standard deviation (*f_ct,sp_*) (MPa]	0.28	0.25	0.24
**Residual tensile strength**			
*LOP* = *f_L_* (MPa]	3.59	4.82	4.67
Increase to NC (%]	-	34	30
Standard deviation (*f_L_*) (MPa]	0.18	0.07	0.38
*f_R1k_* (MPa] at *CMOD_1_* = 0.5 mm	-	2.29	2.45
*f_R2k_* (MPa] at *CMOD_2_* = 1.5 mm	-	1.15	1.66
*f_R3k_* (MPa] at *CMOD_3_* = 2.5 mm	-	0.91	1.26
*f_R4k_* (MPa] at *CMOD_4_* = 3.5 mm	-	0.72	0.96
*f_R3k_*/*f_R1k_* > 0.5	-	0.4 < 0.5	0.51 > 0.5
*f_R1k_*/*f_L_* > 0.4	-	0.47 > 0.4	0.53 > 0.4
**Tensile strength by WST method**			
*σ_NT_* (MPa]	1.79	2.63	3.04
Increase to NC (%]	-	47	70

**Table 6 materials-14-00256-t006:** Results of pull-off peel tests.

Adhesion Strength-Pull-Off Test	NC	SFRC	RSFRC
*f_h_* (MPa]	2.50	2.45	2.49
Standard deviation (*f_h_*)	0.41	0.37	0.46

**Table 7 materials-14-00256-t007:** Results of Boehme’s abrasion resistance tests.

Boehme’s Abrasion Resistance	NC	SFRC	RSFRC
Δ*V* (cm^3^/50 cm^2^]	8.48	7.58	7.47
Standard deviation (Δ*V*)	1.04	0.56	0.38
Δ*l* (mm]	1.70	1.53	1.47
Standard deviation (Δ*l*)	0.17	0.12	0.10
Boehme’s abrasion resistance class	A9	A9	A9

**Table 8 materials-14-00256-t008:** Determination of energy consumption and CO_2_ emissions for 1 m^3^ of fibre concrete with ISF fibres.

Concrete Component	Energy Consumption Factor (MJ/kg]	CO_2_ Emission (kg CO_2_/kg]	Amount in 1 m^3^ (kg]	Energy Consumption (MJ]	CO_2_ Emission (kg CO_2_]
cement (kg]	3 *	0.3 **	320	960	96
water (dm^3^]	0.05 **	0 **	160	8	0
sand 0/2 (kg]	0.1 **	0.007 **	805	80.5	5.635
gravel 2/8 (kg]	0.1 **	0.007 **	684	68.1	4.788
gravel 8/16 (kg]	0.1 **	0.007 **	454	45.6	3.192
fibre ISF (kg]	21.0 ***	1.95 **	25	525	48.75
			**SUMA**	**1687.5** (MJ]	**158.4** (kg CO_2_]

(*) calculation based on indicators published by the Central Statistical Office [24] on the assumption that CEM II/B-S 42.5N-NA contains 70%; (**) indicators derived from publications [25]; (***) calculation based on indicators published by the Central Statistical Office [24] assuming the following processes in the production of steel fibres: production of iron pig iron, steel from electric furnaces, cold rolling of steel and wire drawing. In addition, 0.5 MJ/kg wire cutting energy was adopted.

**Table 9 materials-14-00256-t009:** Determination of energy consumption and CO_2_ emissions for 1 m^3^ of fibre concrete with RSF fibres.

Concrete Component	Energy Consumption Factor (MJ/kg]	CO_2_ Emission (kg CO_2_/kg]	Amount in 1 m^3^ (kg]	Energy Consumption (MJ]	CO_2_ Emission (kg CO_2_]
cement (kg]	3 *	0.3 **	320	960	96
water (dm^3^]	0.05 **	0 **	160	8	0
sand 0/2 (kg]	0.1 **	0.007 **	801	80.1	5.607
gravel 2/8 (kg]	0.1 **	0.007 **	681	68.1	4.767
gravel 8/16 (kg]	0.1 **	0.007 **	454	45.4	3.178
fibre RSF (kg]	0 ****	0 ****	50	0	0
			**SUMA**	**1161.6** (MJ]	**109.55** (kg CO_2_]

(*)–(**) according to Table 8. (****) zero energy consumption and zero CO_2_ emission value were assumed for RSF fibres constituting waste during rubber recovery.

## Data Availability

Not applicable.

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
