# Peer review of "Mechanical Performance and Environmental Assessment of Sustainable Concrete Reinforced with Recycled End-of-Life Tyre Fibres"

_materials, 2021, doi:10.3390/ma14020256_

Round 1

Reviewer 1 Report

The article analysis mechanical performance and assess environmental impact of concrete with recycled tire fibers. While reading the article, I have concluded the following remarks:

1) There are no explanation of what is DIC and KAHL. Additionally, there are no information on suppliers and/or manufacturers of the materials used for the study.

2) As SEM images are quite small, it is hard to see the scaling, therefore, I suggest deleting the black zone under the image and putting the scale bar in the upper right corner so it could be more visible.

3) The scaling and quality of EDS spectrum must be increased.

4) Please increase the scaling of Figure 11, it is hard to see anything in such graphs

5) Please use dots in numerical values instead of commas. Moreover, add upper and lower limits or at least standard deviations of the results obtained in graphs and Tables where appropriate. 

6) Figure 14. Please use the same rounding rule for each results presentation. Why SFRC has Ac of 2.15 while RSFRC and NC  only 1.9 and 1.7 respectively?

7) there is no discussion section or at least Results and discussion section where obtained results and tendencies are discussed with other authors works.

8) Conclusion section must be rewritten. I must contain the most important observations with the results. Now, it only repeats the respective parts of results section.

Author Response

Response to Reviewer 1 Comments

Point 1:  There are no explanation of what is DIC and KAHL. Additionally, there are no information on suppliers and/or manufacturers of the materials used for the study.

Response 1: In regard to reviewer’s remarks the authors have added the full name of the digital image correlation (DIC) method in revised Abstract (in line 21). KAHL processing is additionally explained between lines 58-62, in subsection 2.2 on characteristics of recycled steel fibres (RSF).

Point 2:  As SEM images are quite small, it is hard to see the scaling, therefore, I suggest deleting the black zone under the image and putting the scale bar in the upper right corner so it could be more visible

Response 2: According to the reviewer’s suggestion, the black zone under the image in SEP images  (Figure 2) was deleted and the scale bars in the upper right corner were submitted.

Point 3:  The scaling and quality of EDS spectrum must be increased.

Response 3:   EDS spectrum for RSF recognized under X-ray analysis (Figure 3) was increased. In addition, new table of chemical elements content has been attached at that figure.

Point 4:  Please increase the scaling of Figure 11, it is hard to see anything in such graphs

Response 4: In regard to reviewer’s remarks the authors increased the scaling of Figure 11.

Point 5:  Please use dots in numerical values instead of commas. Moreover, add upper and lower limits or at least standard deviations of the results obtained in graphs and Tables where appropriate. 

Response 5: The authors had checked again the whole article and corrected all numerical values where appropriate. Standard deviations of the results were added in Tables 4-7.

Point 6:  Figure 14. Please use the same rounding rule for each results presentation. Why SFRC has Ac of 2.15 while RSFRC and NC  only 1.9 and 1.7 respectively?

Response 6: The Figure 14 and lines: 282-284 were corrected towards using the same rounding rule (0.00) for air content results (Ac).

Point 7:  There is no discussion section or at least Results and discussion section where obtained results and tendencies are discussed with other authors works.

Response 7: Considering the reviewer’s remarks, authors had decided to split Results and Discussion into separate sections no 4 and 5 respectively. The authors hope that in the corrected Discussion section, the interpretation of results is presented in the broader context of fibre concrete characteristics.

Point 8:  Conclusion section must be rewritten. I must contain the most important observations with the results. Now, it only repeats the respective parts of results section.

Response 8: Conclusions were rewritten. The corrected section includes most important   observations with quantitive information on results, limitations of the study and further recommendations.

Reviewer 2 Report

The manuscript presents original data on an important problem: the recycling of metal used as tire reinforcement in concrete.

Publication is recommended after the following points have been improved.

Add some quantitative information in the abstract: length of steel fibres, added volume in composite materials, mechanical strength, etc.

Introduction, lines 44-47: add references to pneumatic separation. Add the composition of the steel fibres and the mechanical strength.

Lines 72: RSF is defined in the abstract but must also be defined when first used in the text. Ditto for the acronym SFRC and RSFRC, lines 82.

Lines 98-110: It will be better to present the characteristics of the two types of fibres first (Table 2 before Table 1) and add RSF characteristics (average / representative) in the new table 1. The reader should see a clear comparison of the two types of fibres. The authors should explain why the addition of fibre is different for the two categories: 25 and 50 kg / m3?

Fig. 3: increase the contrast and size of the labels; replace the table image with a standard table. Delete K (Kalpha? line) In the list of elements. The quality of Figure 7 is too poor. Get a better quality image or delete the figure.

End of § 3: A table giving the size of the coupon used will be useful.

Figure 1: labels are too small!

Figure 12: add the volume % of fibres (red) and pore (green) for the two samples in the caption.

Figure 13: Increase the contrast.

Table 4: use a dot, not a comma to separate decimals!

Lines 332: the authors should mention that the comparison is made for composite materials with different volume (and mass) of reinforcement.

Figure 16: a zoom on the deflection range 0-0.03 is necessary to discuss the mechanical behaviour: visualization of the linear behaviour, appearance of the difference in linearity (cracks), relation between the length of fibre and the maximum of the curve. It is a pity that a comparison between concrete with the same volume of fibrous reinforcement has not been made and discussed. Figures 19 and 20: add scale.

Figure 22: add a zoom of the 0-0.1 range for a better discussion.

Tables 8 and 9: m3 with 3 as superscript. Layout: try to have both Tables very close (same page) for a clear comparison.

Conclusions: add quantitative information

Author Response

Response to Reviewer 2 Comments

Point 1:  Add some quantitative information in the abstract: length of steel fibres, added volume in composite materials, mechanical strength, etc.

Response 1:    Due to the limited abstract’s length the authors have decided to present the detailed quantitative information on the length of steel fibres, added volume in composite materials and their mechanical strength in section 2.2 (lines 95-160).  The submitted abstract (194 words) gives an overview of the scope of research conducted on mechanical and physical features of the concrete composite with the addition of recycled steel fibres (RSF) in relation to the steel fibre concrete commonly used for industrial floors.

Point 2:  Introduction, lines 44-47: add references to pneumatic separation. Add the composition of the steel fibres and the mechanical strength.

Response 2: The authors had visited several tire recycling plants using ambient method before they started their research. Therefore, the given information on pneumatic separation (lines 44-46) is based on the authors’ experiences and no additional references were submitted in the article.

Point 3:  Lines 72: RSF is defined in the abstract but must also be defined when first used in the text. Ditto for the acronym SFRC and RSFRC, lines 82.

Response 3:   Considering the reviewer’s remarks, authors had added the explanations of the RSF acronym in line 51, and acronyms: NC, SFRC, and RSFRC in corrected section 2.3. (Composition of concrete mix) between lines 162-164.

Point 4:  Lines 98-110: It will be better to present the characteristics of the two types of fibres first (Table 2 before Table 1) and add RSF characteristics (average / representative) in the new table 1. The reader should see a clear comparison of the two types of fibres. The authors should explain why the addition of fibre is different for the two categories: 25 and 50 kg / m3 ?

Response 4: According to the reviewer’s suggestion, the section no 2 on Materials was corrected. RSF and ISF fibre characteristics were demonstrated in the new Table 1, and new Table 2 respectively, both Tables located in the article closer to each other. The explanation on the different fibre content (50 kg/m3  for composites with RSF and 25 kg/m3 for composites with ISF) was additionally explained between the lines 171-173.

Point 5:  Fig. 3: increase the contrast and size of the labels; replace the table image with a standard table. Delete K (Kalpha? line) In the list of elements. The quality of Figure 7 is too poor. Get a better quality image or delete the figure.

Response 5: Due to the reviewer’s remarks, EDS spectrum for RSF recognized under X-ray analysis (Figure 3) wasincreased. In addition, new table of chemical elements content has been attached at that figure. Moreover, Figure 7 was replaced by the better quality image.

Point 6:  End of § 3: A table giving the size of the coupon used will be useful

Response 6: Some information on the number of samples used for abrasion resistance test were added in line 240.

Point 7: Figure 1: labels are too small!

Response 7: In regard to reviewer’s remarks the authors increased the scaling of Figure 11.

Point 8:  Figure 12: add the volume % of fibres (red) and pore (green) for the two samples in the caption.

Response 8: The authors added the information on the volume % of fibres (red) and pore (green) for the two samples at Figure 12.

Point 9:  Figure 13: Increase the contrast.

Response 9: According to reviewer’s suggestion, the authors had increased the contrast in Figure 13.

Point 10:  Table 4: use a dot, not a comma to separate decimals!

Response 10: The authors had checked again the whole article and corrected all numerical values where appropriate.

Point 11:  Lines 332: the authors should mention that the comparison is made for composite materials with different volume (and mass) of reinforcement.

Response 11: The information on the different volume of fibres for different composites is clearly presented in section no 2.3 on composition of concrete mixes. It should be mentioned, that considering the other reviewer’s remarks, authors had decided to split Results and Discussion into separate sections no 4 and 5 respectively. The additional information on the different volume of fibres for different composites is given in Discussion section no 5.2 (lines between 408-410).

Point 12:  Figure 16: a zoom on the deflection range 0-0.03 is necessary to discuss the mechanical behaviour: visualization of the linear behaviour, appearance of the difference in linearity (cracks), relation between the length of fibre and the maximum of the curve. It is a pity that a comparison between concrete with the same volume of fibrous reinforcement has not been made and discussed.

Response 12:  The authors had decided to increase the Figure 16 instead of zooming a part of diagram. The main objective of comparison of diagrams of F-deflection relationships for non-fibre (NC) and fibre (SFRC and RSFRC) concretes was to prove a very high convergence of results for samples reinforced with diffuse ISF and RSF fibres obtained during the 3-point test with deflection up to 0.4 mm. Referring to the fact that half of RSF fibres are not effective in concrete composite due to their irregular shape, a comparison between concrete with the same volume of fibrous reinforcement would evidently show much lower tensile strength of composite reinforced with RSF fibres. Such research will contradict the thesis of the article of the possibility of replacing industrial fibres with RSF fibres without reducing fibre concrete's strength parameters.

Point 13: Figures 19 and 20: add scale.

Response 13: At Figure 19 the colour scale is submitted, however the colour scale indicates only the intensity of the horizontal deformation of the RSFRC sample. The authors had added the scale at Figure 20, according to the reviewer’s suggestion.

Point 14:  Figure 22: add a zoom of the 0-0.1 range for a better discussion.

Response 14: The authors had decided to increase the Figure 22 instead of zooming a part of diagram. The presented discussion refers to the comparison of behaviour of the samples with the RSF and ISF fibres in the elastic zone, which indicates the possibility of replacing ISF fibres with recycled RSF fibres without reducing the strength parameters fibre concrete.

Point 15:  Tables 8 and 9: m3 with 3 as superscript. Layout: try to have both Tables very close (same page) for a clear comparison.

Response 15: According to the reviewer’s suggestion, both Tables 8 and 9 are situated at the same page no 16. Moreover, they were adjusted towards correct superscript submission.

Point 16:   Conclusions: add quantitative information

Response 16: Conclusions were rewritten. The corrected section includes most important   observations with quantitive information on results, limitations of the study and further recommendations.

Round 2

Reviewer 1 Report

I still cannot see the upper and lower limits in bar-type graphs. If you have tested more than one sample for each test, please add the upper and lower limits with 95% confidence interval or at least standard deviations

Author Response

Response to Reviewer 1 Comment

Point 1:  I still cannot see the upper and lower limits in bar-type graphs. If you have tested more than one sample for each test, please add the upper and lower limits with 95% confidence interval or at least standard deviations

Response 1:    Considering the reviewer’s remark, the authors had added the standard deviations in bar-type graphs at Figures 13, 14, and 15 in the Results section. Figure 21, submitted in Discussion section, refers only to the discussion under comparison of three different tensile strength methods.  The results of tensile strength of tested concretes appointed by the Brazilian method, 3-point test and WST are presented in detail incl. standard deviation in Table 5 in the previous Results section.

Reviewer 2 Report

The text has been clarified. A single remark, the abstract, even short, must not only present the objective of the work but also give the main results quantitatively.

Author Response

Response to Reviewer 2 Comment

Point 1:  The abstract, even short, must not only present the objective of the work but also give the main results quantitatively.

Response 1:    Considering the reviewer’s remark, the authors had added the quantitative information in the abstract section on the main results of the tensile strength of the composite between lines 18-21 and the environmental impact between lines 23-24.